# Metagenomic Analysis Reveals the Effects of Different Land Use Types on Functional Soil Phosphorus Cycling: A Case Study of the Yellow River Alluvial Plain

**DOI:** 10.3390/microorganisms12112194

**Published:** 2024-10-30

**Authors:** Ming Wen, Yu Liu, Chaoyang Feng, Zhuoqing Li

**Affiliations:** 1State Key Laboratory of Environmental Benchmarking and Risk Assessment, Chinese Research Academy of Environmental Sciences, Beijing 100012, China; w1255270260@163.com (M.W.); lyu@craes.org.cn (Y.L.); fengchy@craes.org.cn (C.F.); 2College of Ecology, Lanzhou University, Lanzhou 730000, China

**Keywords:** land use type, soil phosphorus cycling, functional genes, macrogenome, ecosystems

## Abstract

Phosphorus (P) is a crucial limiting nutrient in soil ecosystems, significantly influencing soil fertility and plant productivity. Soil microorganisms adapt to phosphorus deficiency and enhance soil phosphorus effectiveness through various mechanisms, which are notably influenced by land use practices. This study examined the impact of different land use types (long-term continuous maize farmland, abandoned evolving grassland, artificial tamarisk forests, artificial ash forests, and wetlands) on soil phosphorus-cycling functional genes within the Tanyang Forest Farm in a typical region of the Yellow River alluvial plain using macro genome sequencing technology. The gene cluster related to inorganic phosphorus solubilization and organic phosphorus mineralization exhibited the highest relative abundance across different land use types (2.24 × 10^−3^), followed by the gene cluster associated with phosphorus transport and uptake (1.42 × 10^−3^), with the lowest relative abundance observed for the P-starvation response regulation gene cluster (5.52 × 10^−4^). Significant differences were found in the physical and chemical properties of the soils and the relative abundance of phosphorus-cycling functional genes among various land use types. The lowest relative abundance of soil phosphorus-cycling functional genes was observed in forestland, with both forestland types showing significantly lower gene abundance compared to wetland, farmland, and grassland. Correlation analysis and redundancy analysis (RDA) revealed a significant relationship between soil physicochemical properties and soil phosphorus-cycling functional genes, with ammonium nitrogen, organic carbon, total nitrogen, and pH being the main environmental factors influencing the abundance of these genes, explaining 70% of the variation in their relative abundance. Our study reveals land use’s impact on soil phosphorus-cycling genes, offering genetic insights into microbial responses to land use changes.

## 1. Introduction

Phosphorus (P), a crucial element for sustaining soil ecosystem balance, is regarded as the second most limiting nutrient in soil ecosystems after nitrogen [1]. The phosphorus cycle, a fundamental depositional cycle, primarily exists in organic and inorganic forms. The patterns of phosphorus storage and transformation directly affect nutrient acquisition and utilization by organisms, thus impacting soil ecosystem functions and services [2]. Examining the mechanisms of soil phosphorus transformation and regulation is essential for optimizing soil ecosystem phosphorus resources and improving soil phosphorus effectiveness.

In recent years, human activities have increasingly impacted the natural ecological environment of the Yellow River alluvial plain, significantly affecting the landscape and leading to considerable shifts in land use patterns [3]. There has been a widespread reduction in wetland areas and the clearance of native vegetation to expand agricultural and pastoral lands. These land use changes are associated with modifications in soil physical, chemical, and biological processes due to various anthropogenic management practices, such as tillage, cultivation, and fertilization, which in turn alter the distribution of soil phosphorus (P) forms [4,5,6,7]. Research consistently shows that different land use types can significantly modify soil properties, including texture, pH, carbon content, and nutrient composition [8,9,10,11]. These modifications lead to changes in soil microbial communities, which are crucial in numerous soil biochemical reactions and significantly influence soil nutrient dynamics and ecosystem processes. Microbial communities are key drivers of soil P turnover and biogeochemical cycling. For example, under phosphorus deficiency, microorganisms may secrete phosphatases via the phosphate starvation (Pho) regulon to solubilize and access bioavailable orthophosphate [12]. Extensive studies have highlighted the significant impact of land use type on specific functional genes involved in phosphorus cycling, such as the alkaline phosphatase gene (*phoA*) and phytase gene (*appA*), across various global regions and climatic zones [13,14]. Andrew et al. [15] observed an increased abundance of the *phoA*, *phoD*, and *phoX* genes in bare fallow soils compared to grasslands and arable lands in the UK. Kathia et al. [16] found that phosphatase genes had the lowest abundance in scrub and agricultural fields in their study of soil bacterial communities across five land use types in the Mezquital Valley, Mexico. Wu et al. [17] assessed the effects of four different types of wetland degradation conditions on soil microbial communities in the Sanjiang Plain and found that wetland degradation led to a decrease in soil nutrients and a decrease in the abundance of dominant phyla in the soil bacterial and fungal communities. Li et al. [18] found that the introduction of broadleaf species not only increased soil nutrient content but also had a significant effect on the increase in the diversity of soil fungal communities, resulting in a more diverse microbial community in mixed forests. However, most research has focused on the impact of land use on individual functional genes within the phosphorus cycle (e.g., the alkaline phosphatase gene *phoD*) or multiple genes within a single process (e.g., the mineralization of soil phosphates). A more comprehensive understanding of how land use changes affect multiple genes across various processes in the soil phosphorus cycle is required.

This study examined the impact of different land use types (long-term continuous maize farmland, abandoned evolving grassland, artificial tamarisk forests, artificial ash forests, and wetlands) on soil phosphorus-cycling functional genes within the Tanyang Forest Farm in a typical region of the Yellow River alluvial plain using macro genome sequencing technology. The research aimed to analyze variations in the relative abundance of 33 functional genes associated with seven key processes within the soil phosphorus cycle across these land use types. Alongside laboratory analyses, the study identified changes in the fundamental physicochemical properties of soils associated with different land use types. The objectives of the study were threefold: (1) to assess the impact of land use on the relative abundance of functional gene groups involved in the seven main processes of the soil phosphorus cycle; (2) to investigate differences in the abundance and composition of the 33 functional genes of the soil phosphorus cycle among various land use types; and (3) to identify the key environmental factors regulating the functional genes of the soil phosphorus cycle in response to land use changes. We hypothesized that wetlands and grasslands with richer vegetation and biomass would result in high levels of abundance of functional phosphorus-cycling genes.

## 2. Materials and Methods

### 2.1. Study Area

The study area was carefully selected within Tanyang Forest Farm, Wudi County, Binzhou City, Shandong Province (coordinates: 37°53′ N, 117°56′ E), located at the northernmost edge of Shandong Province within the Yellow River alluvial plain (Figure 1). The predominant soil type in this region is coastal tidal soil. The climate is classified as a continental monsoon climate of the northern temperate East Asian zone, characterized by four distinct seasons and a marked contrast between wet and dry periods. The region has an average annual temperature of 13.4 °C and an average annual precipitation of 624.9 mm.

This investigation encompassed five distinct land use types: tamarisk forest, ash forest, wetland, farmland, and grassland. These land use types were located in close proximity to each other, with uniform slopes and tidal soils, and lacked topographic differences. The adjacency and historical continuity of these land use types, each exceeding a decade, ensured a high level of uniformity in variables other than land use, such as climate, topography, and parent material. The agricultural land was used for maize (*Zea mays*) cultivation, with a history of 10 years of continuous cropping without fertilizer application. The wetland was dominated by reed species (Phragmites australis) from the Gramineae family. The grassland, resulting from 6 years of land abandonment, featured dominant species, including dogwood (*Setaria viridis*) and ryegrass (*Lolium perenne* L.), both from the Gramineae family. The tamarisk forest, established in 2013, was dominated by *Tamarix chinensis* (*Tamaricaceae* family), while the ash forest, established in 2009, was dominated by Fraxinus chinensis (*Oleaceae* family). None of the five land use types has a history of artificial fertilizer application in the last 10 years.

### 2.2. Soil Sample Collections

Soil samples were collected from five replicate plots for each land use type. The plots for ash and tamarisk forests measured 10 m × 10 m, while the wetland, farmland, and grassland plots each covered an area of 1 m × 1 m. The plot sizes for ash and tamarisk forests were set larger to cover a wider area and thus better reflect the soil properties of forest ecosystems. In contrast, wetlands, agricultural land, and grassland plot sizes were designed to be smaller, taking into account the fact that the soil properties of these land use types are likely to have a higher degree of homogeneity at smaller scales. A five-point sampling method was employed, with samples collected from five discrete points within each plot and combined into a composite sample, targeting the 0–20 cm depth of the surface soil. A total of 25 soil samples were collected. This strategy of mixing samples helps to balance possible small-scale spatial variability. It ensures that the soil samples from each sample site are a true reflection of the average soil condition of its land use type. Following collection, roots and large stones were removed from the samples, which were then placed in sterile plastic bags, preserved in ice boxes, and transported to the laboratory. In the laboratory, the soil samples were sifted through a 2.0 mm sieve, resulting in two separate portions: one portion was air-dried at room temperature for the analysis of fundamental soil physicochemical properties, while the other was stored at an ultralow temperature (–80 °C) for future analysis of key functional genes involved in soil microbial phosphorus cycling.

### 2.3. Determination of Physical and Chemical Properties

The following soil physicochemical properties were assessed: soil pH, electrical conductivity (EC), total nitrogen (TN), ammonium nitrogen (NH_4_^+^-N), total potassium (TK), available potassium (AK), soil organic carbon (SOC), and total phosphorus (TP). Soil pH was measured using the glass electrode method with a soil–water ratio of 1:2.5 [19]. Conductivity was determined using the electrode method [20]. Ammonium nitrogen content was quantified using the indophenol blue colorimetric method [21]. Total nitrogen content was assessed by the semi-micro Kjeldahl method [22]. Total potassium content was measured using hydrofluoric acid–perchloric acid digestion followed by flame photometry [23]. Available potassium content was determined by the ammonium acetate leaching-flame photometric method [24]. Soil organic carbon content was measured using external heating with potassium dichromate [25]. Total phosphorus content was quantified via a colorimetric method using a UV spectrophotometer following digestion with concentrated sulfuric and perchloric acids [26].

### 2.4. Determination of Functional Genes Involved in Soil Phosphorus Cycling

In this study, DNA extraction from soil samples was performed using the Fast DNA SPIN Kit for Soil (MP Bio, Santa Ana, CA, USA). Post-extraction, DNA purity and integrity were assessed through agarose gel electrophoresis (AGE). DNA samples meeting quality standards were fragmented to an approximate size of 350 base pairs using a Covaris ultrasonic disruptor(Covaris, Woburn, MA, USA). Library preparation involved end repair, A-tail addition, adapter ligation, purification, and PCR amplification. The effective concentration of the library was measured using qPCR, ensuring it exceeded 3 nM [27]. Sequencing was carried out on the Illumina PE150 platform by Novogene Bioinformatics Technology Co., Ltd. (Beijing, China). The resulting data were assembled and analyzed using the MEGAHIT software (V1.2.9) suite, with open reading frame (ORF) prediction performed using MetaGeneMark [28]. Sequence dereplication was accomplished with CD-HIT software (V4.8.1). Gene abundance was determined by correlating read counts with gene lengths. Unigenes were aligned with the KEGG database using DIAMOND software (V4.6.8) to assign corresponding KEGG functions [29], and functional category abundance was determined by aggregating gene abundance associated with KO identifiers. The phosphorus cycle functional gene database compiled in this study was based on a review of the published literature and the SWISS-PROT database, now included in the UniProt review section. This resource was utilized to validate and enhance the gene names, functional attributes, and EC numbers of relevant enzymes. It served as the basis for functional gene sequence comparisons and the filtering of comparative results, ensuring consistency across gene names, KOs, and functional descriptions. This study examined phosphorus cycle functional genes related to phosphorus activation, uptake and transport, and P-starvation response regulation (Table 1). The phosphorus-related genes were categorized into three functional groups: phosphate ester mineralization, phosphonate mineralization, and inorganic phosphate solubilization. Genes related to phosphorus uptake and transport included those involved in phosphonate transport, phosphate transport, and inorganic phosphate transport. P-starvation response regulatory genes were identified as phosphate-regulated functional genes [30].

### 2.5. Statistical Analysis

Data analysis was performed using SPSS software version 27. Due to the non-normal distribution of functional genes, the Kruskal–Wallis nonparametric test was employed to evaluate significant differences between land use types [31]. This method is based on rank order rather than mean value, which requires less normality of the data and is suitable for the characteristics of the data in this study. Soil physicochemical properties were analyzed using analysis of variance (ANOVA), and multiple comparisons were conducted using Duncan’s method, with significance determined at *p* < 0.05 [32]. Correlations between key functional genes of the phosphorus cycle and soil physicochemical properties were analyzed using Spearman correlation analysis [33]. The relationships between the abundance of phosphorus-cycling functional genes and soil physicochemical properties were assessed through redundancy analysis (RDA) and visualized using CANOCO5 software (V5.1.2) [34]. The relative abundance of functional genes was plotted using GraphPad 9.5 software. Mantel test correlation analysis was conducted using the “vegan” package in R software (V2.6-8), with correlations between variables visualized using ChiPlot (www.chiplot.online).

## 3. Results

### 3.1. Differences in Soil Physicochemical Properties Across Land Use Types

The study found significant differences in soil physicochemical properties among different land use types (Table 2). The pH values of wetland and grassland soils were considerably higher compared to those of other land uses, with the lowest pH values found in farmland soils. The electrical conductivity (EC) of wetland soils was markedly higher than that of the other four land use types. Regarding potassium content, both total and available (AK), tamarisk forest soils exhibited the highest levels, while wetland soils had the lowest available potassium (AK) levels. The total nitrogen (TN) content in ash forest and tamarisk forest soils was significantly higher than that in farmland, wetland, and grassland soils. Additionally, tamarisk forest soils contained a substantially higher concentration of ammonium nitrogen (NH_4_^+^-N) compared to other land uses. Conversely, the ammonium nitrogen content was significantly lower in grassland soils compared to farmland, wetland, tamarisk forest, and ash forest soils. The total phosphorus (TP) content was highest in grassland soils, whereas ash forest soils displayed a notably lower TP content. The organic carbon (SOC) content in wetland soils was significantly higher than that in tamarisk forest, grassland, ash forest, and farmland soils.

### 3.2. Functional Gene Composition of Soil Phosphorus Cycling in Different Land Use Types

The distribution of phosphorus-cycling functional genes across the five land use types in the Yellow River alluvial plain showed a generally consistent pattern (Figure 2). The cluster for inorganic phosphorus solubilization functional genes exhibited the highest relative abundance (1.19 × 10^−3^), while the phosphate transport functional gene cluster had the lowest relative abundance (3.44 × 10^−4^). The genes with the highest relative abundance in the phosphate mineralization, phosphonate mineralization, and inorganic phosphate solubilization functional gene clusters were *glpK* (2.13 × 10^−4^), *phnJ* (9.37 × 10^−6^), and *ppk* (4.14 × 10^−4^), respectively. For the phosphonate transport, phosphate transport, and inorganic phosphate transport functional gene clusters, the genes with the highest relative abundance were *phnE* (1.05 × 10^−4^), *ugpB* (9.20 × 10^−5^), and *pstB* (3.06 × 10^−4^). In the phosphate regulation functional gene cluster, *phoR* had the highest relative abundance at 2.56 × 10^−4^.

### 3.3. Differences in the Relative Abundance of Functional Soil Phosphorus-Cycling Genes Across Land Use Types

The functional gene clusters related to the seven primary processes of the soil phosphate cycle—phosphate mineralization, phosphonate mineralization, inorganic phosphorus solubilization, phosphonate transport, phosphate transport, inorganic phosphate transport, and phosphate regulation—demonstrated significant variation across different land use types (Figure 3). Notably, the gene cluster for soil phosphate mineralization was most abundant in agricultural soils and least abundant in ash forest soils. In wetland soils, the gene clusters for phosphonate mineralization and inorganic phosphate solubilization were most prevalent, whereas these clusters were at their lowest in white wax forest soils. The gene clusters associated with phosphonate transport, phosphate transport, and inorganic phosphate transport were also most abundant in wetland soils. On the other hand, tamarisk forest soils had the lowest relative abundance for both phosphate transport and inorganic phosphate transport gene clusters, while white wax forest soils had the lowest values for phosphonate transport. The phosphate regulation gene cluster was most prevalent in grassland soils, significantly exceeding its abundance in other land use types (artificial tamarisk forests, artificial ash forests, farmland, and wetlands) (*p* < 0.05). Conversely, tamarisk forest soils exhibited the lowest relative abundance for this gene cluster. Overall, the abundance of phosphorus-cycling functional genes was highest in wetland soils and lowest in white wax forest soils.

The relative abundance of soil phosphorus-cycling genes exhibited significant variation across different land use types (Figure 4). Genes involved in soil inorganic phosphorus solubilization and organic phosphorus mineralization are classified into three functional types: phosphate mineralization, phosphonate mineralization, and inorganic phosphorus solubilization. Within the phosphate mineralization genes, the *glpK* gene showed the highest relative abundance, while the *phoN* gene had the lowest relative abundance across the five land use types. Specifically, the relative abundance of both *glpK* and *phoN* genes was significantly higher in wetland soils compared to other land use types (*p* < 0.05). For phosphonate mineralization genes, the *phnW* gene exhibited the highest relative abundance, whereas the *phnA* gene had the lowest relative abundance. The *phnW* gene was notably more abundant in wetland soils than in other land use types, while the *phnA* gene was most prevalent in grassland soils.

In the case of inorganic phosphorus solubilization genes, the *ppk* gene had the highest relative abundance, with the *pqqE* gene having the lowest relative abundance. The relative abundances of the *ppk* and *pqqE* genes were significantly greater in agricultural and wetland soils, respectively, than in the other land use types (*p* < 0.05). Phosphorus uptake- and transport-related genes, which include phosphonate transport, phosphate transport, and inorganic phosphate transport functional genes, presented the highest and lowest relative abundances of the *phnE* and *phnK* genes, respectively. Both the *phnE* and *phnK* genes were significantly more abundant in the grassland soils than in the other land use types (*p* < 0.05). Among the phosphonate transport genes, the *ugpB* gene presented the highest relative abundance, whereas the *glpP* gene presented the lowest relative abundance. Intriguingly, the *glpP* gene presented a significantly greater relative abundance in the tamarisk forest soil than in the wetland and grassland soils (*p* < 0.05). Among the inorganic phosphate transport genes, the *pstB* and *pit* genes presented the highest and lowest relative abundances, respectively, with both being most abundant in agricultural soils. The *phoR* gene, associated with phosphate-regulating functions, presented the highest relative abundance, with grassland soils showing a significantly greater relative abundance than other land use types (*p* < 0.05).

### 3.4. Relative Abundance of Soil Phosphorus-Cycling Functional Genes in Relation to Soil Physicochemical Properties

The Mantel test was employed to evaluate the relationships between key functional gene clusters involved in soil microbial phosphorus cycling and various environmental factors across different land use types. The results revealed that the relative abundance of these gene clusters varied in response to soil environmental factors (Figure 5). Specifically, the abundance of phosphorus-related genes showed a positive correlation with soil organic carbon (SOC). Additionally, the abundance of phosphorus uptake and transport genes was significantly correlated with soil electrical conductivity (EC) and organic carbon (SOC). These genes also exhibited more pronounced positive correlations with soil total potassium (TK) and available potassium (AK) and highly significant positive correlations with soil total nitrogen (TN) and ammonium nitrogen (NH_4_^+^-N). The relative abundance of the P-starvation response regulatory gene cluster was positively correlated with pH, soil total nitrogen (TN), and ammonium nitrogen (NH_4_^+^-N).

The relationships between soil phosphorus functional genes and soil properties were further investigated using RDA, where the eigenvalues of the first and second axes were 33.28% and 13.59%, respectively (Figure 6). These axes together accounted for 46.87% of the total variance, elucidating the variations in soil phosphorus-cycling functional genes across different land use types. Notably, NH_4_^+^-N, SOC, TN, and pH showed significant correlations with the key functional genes involved in soil phosphorus cycling, with NH_4_^+^-N accounting for the largest percentage of the variation (17.4%).

## 4. Discussion

Wetland soils are recognized for their crucial role in carbon sequestration [35]. In this study, the soil organic carbon (SOC) content in wetland soils was significantly higher compared to other land use types, consistent with previous research [36,37]. Grassland, agricultural, and woodland soils, which have lower O_2_ levels compared to wetlands, exhibited reduced SOC contents, potentially due to enhanced nutrient mineralization. Grasslands had notably higher SOC content than farmlands, likely due to intense anthropogenic activities in corn farmland habitats and the impact of unvegetated areas on soil structure, which accelerated organic matter decomposition [38,39]. Tamarisk and ash forests had higher total nitrogen (TN) and ammonium nitrogen (NH_4_^+^-N) contents compared to grassland and farmland soils, aligning with findings by Gelaw et al. [40]. This can be attributed to forests’ ability to transfer nitrogen to the soil through root exudates and surface plant residues, thus enhancing soil microbial activity during growth and development [41]. These results suggest that forested lands are more conducive to nutrient accumulation than agricultural lands and grasslands. Conversely, the total phosphorus (TP) content in agricultural land was lower than that in grasslands, contrary to the findings of Wang et al. [42]. This discrepancy may be due to the long-term fertilization practices in the agricultural land studied by Wang et al., whereas the agricultural land in this study did not receive phosphorus fertilizers during continuous maize cultivation, resulting in greater soil phosphorus depletion.

This study assessed the response of soil phosphorus cycle functional gene cluster abundances to land use changes and found that gene clusters involved in inorganic phosphorus solubilization and organic phosphorus mineralization exhibited the highest relative abundances. These were followed by gene clusters associated with phosphorus transport and uptake, while the P-starvation response regulator gene clusters had the lowest relative abundance. These observations suggest that inorganic phosphorus solubilization and organic phosphorus mineralization are the most critical and complex processes within the phosphorus cycle [30]. These processes encompass the mineralization of phosphates and phosphonates, as well as the solubilization of inorganic phosphorus. The results indicate that the abundance of functional gene clusters related to phosphate mineralization was significantly higher in agricultural soils compared to grasslands and woodlands. This is consistent with Liu et al. [43], who reported the highest total abundance of genes encoding phosphodiesterase and phytase in agricultural fields in southwestern Saskatchewan, Canada, likely due to the influence of crop root exudates. In maize-growing areas, rhizosphere microorganisms such as *Nocardioides* sp. and *Sphingomonas* sp. are critical for the high abundance of functional genes involved in soil phosphate mineralization (*phoD*) as they secrete phosphatases and other enzymes that facilitate the mineralization of organophosphorus compounds [44]. Furthermore, the microbial community composition in maize farmlands may shift due to long-term cultivation, with increased abundances of Ascomycetes, Actinobacteria, and Thick-walled Bacteria, potentially correlating with the high abundance of phosphonate mineralization functional genes [45]. The relative abundance of the phosphonate mineralization functional gene cluster was significantly higher in wetland soils than in other land use types, likely due to the unique hydrological and physicochemical conditions of wetlands, such as periodic flooding and drought, which promote organic carbon accumulation and provide a nutrient-rich environment for microorganisms, thereby enhancing the expression and activity of phosphonate mineralization functional genes [46].

Conversely, the number of functional gene groups associated with inorganic phosphorus solubilization and organic phosphorus mineralization was lower in tamarisk and ash forests compared to grasslands. This observation is consistent with the results reported by Siles et al. [13] and is hypothesized to be due to the reduced biomass and vegetation cover in forested areas relative to grasslands. Grasslands, characterized by their unique vegetation, may contribute more plant residues to the soil, thereby enhancing the soil’s phosphorus supply capacity and increasing the presence of functional genes related to inorganic phosphorus solubilization and organic phosphorus mineralization [47]. A significant positive correlation was observed between the relative abundance of these functional gene clusters and the organic carbon content, which is in line with previous studies [48]. This correlation may be attributed to organic carbon serving as an energy and carbon source for microorganisms, which in turn promotes microbial community growth and metabolic activities that affect phosphorus dissolution and mineralization processes [49].

The phosphorus uptake and transport processes in soil encompass phosphonate transport, phosphate transport, and inorganic phosphate transport. The findings reveal that the relative abundance of functional gene groups involved in these transport mechanisms was notably lower in ash and tamarisk forests compared to wetlands. This difference is hypothesized to result from the rhizosphere’s role in enhancing microbial activity within wetland ecosystems. Vegetation in these ecosystems is known to affect the microbial composition and the expression of functional genes in the rhizosphere soil through root exudates, thereby impacting phosphorus uptake and transport processes. In wetlands, dominant species such as reeds release a variety of organic compounds into the environment. A significant number of microorganisms colonize the extensive root surface area, affecting the volume of root exudates. Rhizomes simultaneously facilitate oxygen transport, enhancing and maintaining substrate hydraulic transport, which influences residence time and creates a favorable environment for microbial growth. This environment supports phosphorus transformation, as well as its uptake and transport by plants. At the genomic level, this is reflected in the high abundance of functional genes related to phosphorus cycling [50,51]. Additionally, the soil phosphorus uptake and transport functional gene cluster showed a highly significant positive correlation with total nitrogen (TN), consistent with the findings of Hu et al. [52]. This correlation is largely attributed to the interdependent relationship between nitrogen and phosphorus, essential nutrients for plant growth, where the availability of one nutrient can affect the demand for the other. Nitrogen inputs may enhance plant requirements for phosphorus, thereby increasing the expression of functional genes associated with phosphorus uptake and transport by soil microorganisms [47].

P-starvation response regulation is a global regulatory mechanism essential for managing inorganic phosphorus. This mechanism includes an endosomal histidine kinase sensor protein encoded by the *phoR* gene, a cytoplasmic transcriptional response regulator encoded by the *phoB* gene, and the metal-binding protein *phoU*, which is encoded by the *phoU* gene [12]. The investigation revealed that the relative abundance of gene clusters related to P-starvation response regulation was significantly higher in grassland soils compared to cultivated farmlands. Among these genes, the *phoR* gene exhibited the highest relative abundance, aligning with findings reported by Liang et al. [53]. Conversely, Siles et al. observed a higher abundance of genes associated with P-starvation response regulation, particularly *phoR* genes, in farmland [13]. This discrepancy may be attributed to differences in climate and land management practices. Generally, the *phoU* and *phoR* genes are more prevalent in phosphorus-rich soils [54]. The study area of Siles et al., which involved North American farmland with a history of long-term fertilization, had adequate phosphorus levels. In contrast, the farmland in this study, characterized by prolonged maize cultivation without fertilization, faced phosphorus loss and depletion, as indicated in Table 1. This resulted in a lower relative abundance of P-starvation response regulatory genes compared to grassland. Dai et al. [55] found that the long-term application of phosphorus fertilizers decreased the relative abundance of the *phoR* gene in farmland, seemingly conflicting with these results. Due to the lack of a significant correlation between macroeconomic data and soil total phosphorus (TP) content in this study, a comprehensive investigation into the relationship between soil phosphorus levels and the abundance of P-starvation response regulation genes was not feasible. Further research is needed to clarify these relationships.

In this study, we investigated the relationship between soil phosphorus-cycling functional genes and various land use types within the Tanyang Forest Farm, situated in Wudi County of the Yellow River alluvial plain. Our research yields valuable insights into the responsiveness of soil phosphorus-cycling functional genes to land use alterations. Nonetheless, the findings are bounded by certain limitations that warrant acknowledgment. Firstly, the geographical scope of our study is confined to a specific sector of the Yellow River floodplain, which may restrict the broader applicability of our conclusions. The diversity of climatic conditions, soil classifications, and vegetation types across different regions could lead to distinct patterns in soil phosphorus cycling and microbial responses that were not captured in our study. Secondly, while we concentrated on five predominant land use types, this selection does not encompass the entire spectrum of land use practices. This limitation may have influenced our comprehensive evaluation of the functional genetic diversity within the soil phosphorus cycle. A more inclusive approach to land use types would be beneficial for future research. To expand the implications of our findings, future studies should contemplate a broader array of geographical and environmental settings. Such investigations will be instrumental in validating our results and in expanding the understanding of soil phosphorus dynamics under varied land management strategies.

## 5. Conclusions

In this study, soil macroeconomics were utilized to investigate the impact of different land use types on the relative abundance of genes related to inorganic phosphorus solubility, organic phosphorus mineralization, phosphorus uptake, and phosphorus transport, as well as regulatory genes responding to phosphorus deficiency within a typical area of the Yellow River alluvial plain. The following conclusions were drawn:

1. Significant differences in the relative abundance of functional genes involved in soil phosphorus cycling were observed across various land use types, with these differences primarily attributed to changes in the underlying soil physicochemical properties.

2. The relative abundance of phosphorus-cycling functional genes in tamarisk and ash forest soils was consistently and significantly lower compared to that in the other land use types. Since the two types of woodland are plantations, their vegetation biomass and cover are lower than that of farmland, grassland, and wetland, so their overall microbial growth and metabolism involved in soil phosphorus-cycling functions are weaker. This is consistent with our hypothesis.

3. A significant correlation was found between key functional genes involved in soil phosphorus cycling and soil physicochemical properties. Soil ammonium nitrogen, organic carbon, total nitrogen, and pH were identified as crucial factors affecting the abundance of phosphorus-cycling functional genes.

This study provides molecular-level data to support the investigation of land use impacts on soil phosphorus cycling. The results enhance the understanding of the interaction between land use and various soil phosphorus-cycling processes and offer essential scientific insights for optimizing land management strategies in the northwestern Shandong section of the Yellow River alluvial plain, with the aim of sustaining and improving soil health.

## Figures and Tables

**Figure 1 microorganisms-12-02194-f001:**
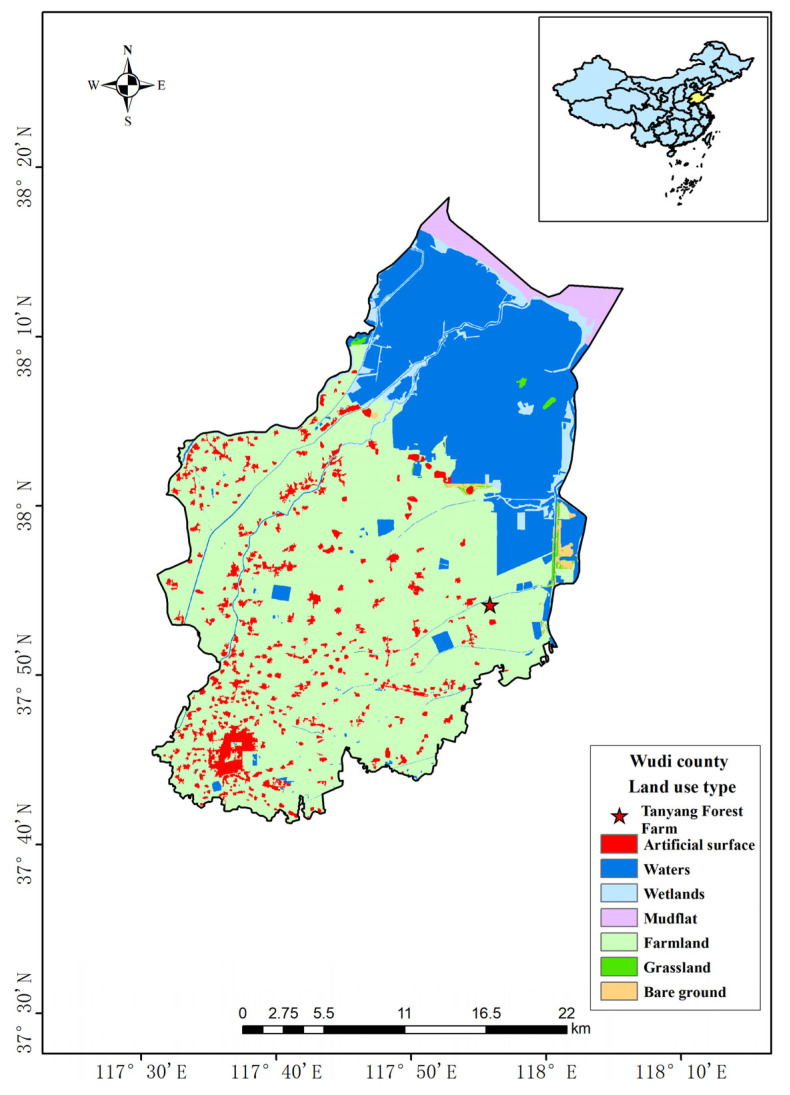
Location of study area.

**Figure 2 microorganisms-12-02194-f002:**
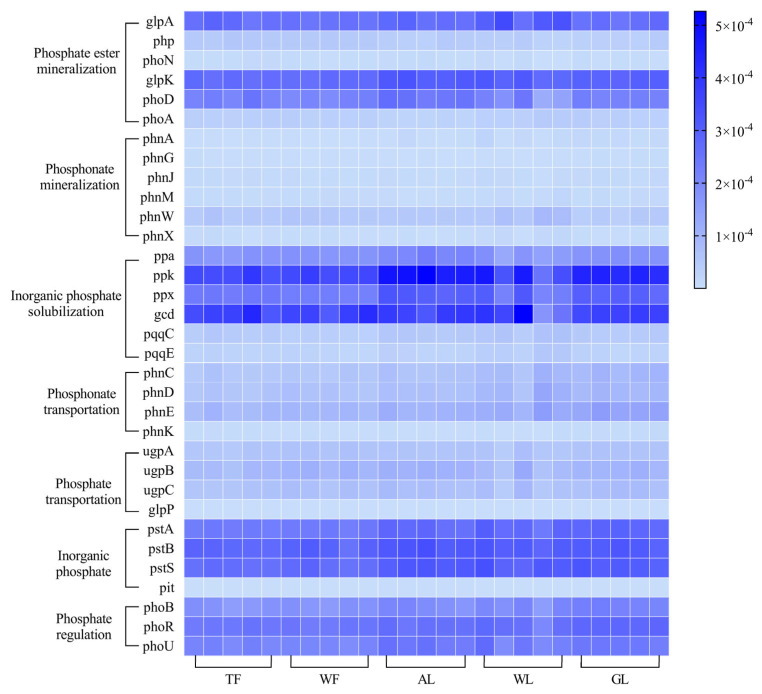
Functional gene composition of soil phosphorus-cycling genes in different land use types. TF: tamarisk forest; WF: white wax forest; AL: agricultural land; WL: wetlands; GL: grassland.

**Figure 3 microorganisms-12-02194-f003:**
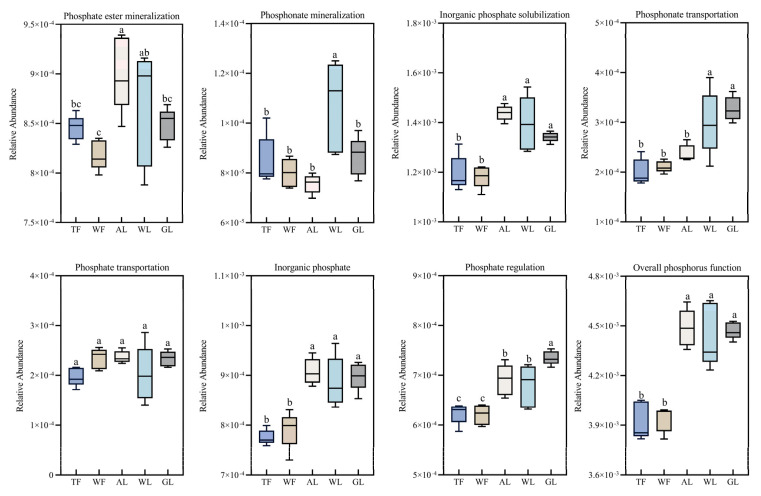
Relative abundance of phosphorus-cycling gene clusters at the functional level in different land use types. Different small letters indicate significant differences at the 0.05 level (ANOVA). TF: tamarisk forest; WF: white wax forest; AL: agricultural land; WL: wetlands; GL: grassland.

**Figure 4 microorganisms-12-02194-f004:**
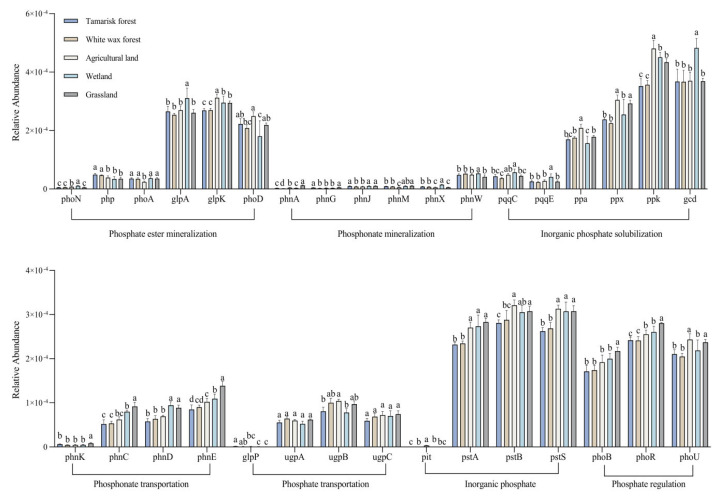
Relative abundance of genes associated with soil phosphorus cycling in different land use types. Different small letters indicate significant differences at the 0.05 level (ANOVA).

**Figure 5 microorganisms-12-02194-f005:**
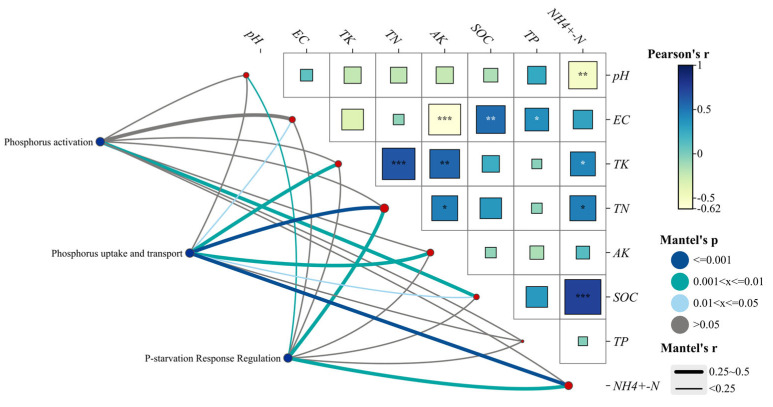
Mantel test of soil phosphorus-cycling functional gene clusters and soil environmental factors. *, ** and *** respectively indicate *p* < 0.05, *p* < 0.1 and *p* < 0.001.

**Figure 6 microorganisms-12-02194-f006:**
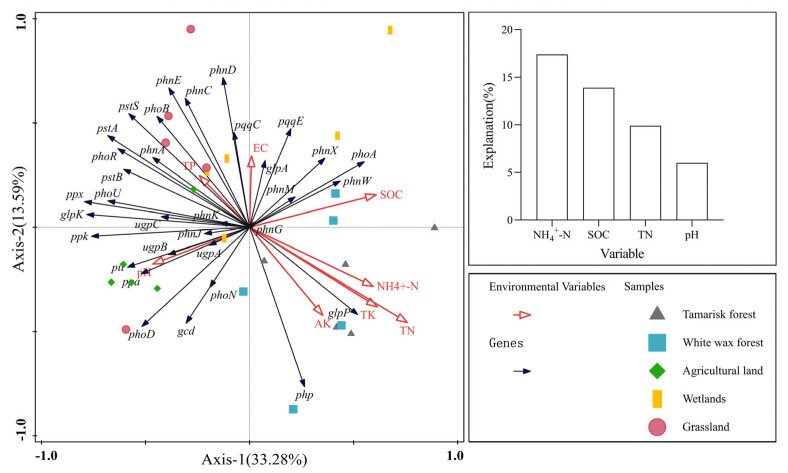
RDA analysis of soil phosphorus-cycling functional genes and soil physicochemical properties.

**Table 1 microorganisms-12-02194-t001:** Analysis of functional phosphorus cycle genes.

KO Number	Gene	Product Name	Functional Gene Grouping	Function in the Nitrogen Cycle
K00111	*glpA*	Glycerol-3-phosphate dehydrogenase	Phosphate ester mineralization	Phosphorus activation
K07048	*php*	Phosphotriesterase-related protein	Phosphate ester mineralization	Phosphorus activation
K09474	*phoN*	Acid phosphatase (class A)	Phosphate ester mineralization	Phosphorus activation
K01077	*phoA*	Alkaline phosphatase	Phosphate ester mineralization	Phosphorus activation
K00864	*glpK*	JNMGlycerol kinase	Phosphate ester mineralization	Phosphorus activation
K01113	*phoD*	Alkaline phosphatase D	Phosphate ester mineralization	Phosphorus activation
K06193	*phnA*	Phosphonoacetate hydrolase	Phosphonate mineralization	Phosphorus activation
K06166	*phnG*	Alpha-D-ribose 1-methylphosphonate 5-triphosphate synthase subunit	Phosphonate mineralization	Phosphorus activation
K06163	*phnJ*	α-D-ribose 1-methylphosphonate 5-phosphate C-P-lyase	Phosphonate mineralization	Phosphorus activation
K06162	*phnM*	Alpha-D-ribose 1-methylphosphonate 5-triphosphate diphosphatase	Phosphonate mineralization	Phosphorus activation
K03430	*phnW*	2-aminoethylphosphonate-pyruvate transaminase	Phosphonate mineralization	Phosphorus activation
K05306	*phnX’*	Phosphonoacetaldehyde hydrolase	Phosphonate mineralization	Phosphorus activation
K01507	*ppa*	Inorganic pyrophosphatase	Inorganic phosphate solubilization	Phosphorus activation
K00937	*ppk*	Polyphosphate kinase	Inorganic phosphate solubilization	Phosphorus activation
K01524	*ppx*	Exopolyphosphatase	Inorganic phosphate solubilization	Phosphorus activation
K00117	*gcd*	Quinoprotein glucose dehydrogenase	Inorganic phosphate solubilization	Phosphorus activation
K06137	*pqqC*	Pyrroloquinoline-quinone synthase	Inorganic phosphate solubilization	Phosphorus activation
K06139	*pqqE*	PqqA peptide cyclase	Inorganic phosphate solubilization	Phosphorus activation
K02041	*phnC*	Phosphonate transport system ATP-binding protein	Phosphonate transportation	Phosphorus uptake and transport and transport
K02044	*phnD*	Phosphonate transport system substrate-binding protein	Phosphonate transportation	Phosphorus uptake and transport and transport
K02042	*phnE*	Phosphonate transport system permease protein	Phosphonate transportation	Phosphorus uptake and transport and transport
K05781	*phnK*	Phosphonate transport system ATP-binding protein	Phosphonate transportation	Phosphorus uptake and transport and transport
K05814	*ugpA*	Sn-glycerol 3-phosphate transport system permease protein	Phosphate transportation	Phosphorus uptake and transport and transport
K05813	*ugpB*	Sn-glycerol 3-phosphate transport system substrate-binding protein	Phosphate transportation	Phosphorus uptake and transport and transport
K05816	*ugpC*	Sn-glycerol 3-phosphate transport system ATP-binding protein	Phosphate transportation	Phosphorus uptake and transport and transport
K02443	*glpP*	Glycerol uptake operon anti-terminator	Phosphate transportation	Phosphorus uptake and transport and transport
K02038	*pstA*	Phosphate transport system permease protein	Inorganic phosphate	Phosphorus uptake and transport and transport
K02036	*pstB*	Phosphate transport system ATP-binding protein	Inorganic phosphate	Phosphorus uptake and transport and transport
K02040	*pstS*	Phosphate transport system substrate-binding protein	Inorganic phosphate	Phosphorus uptake and transport and transport
K16322	*pit*	Low-affinity inorganic phosphate transporter	Inorganic phosphate	Phosphorus uptake and transport and transport
K07657	*phoB*	Phosphate regulon response regulator	Phosphate regulation	P-starvation response regulation
K07636	*phoR*	Phosphate regulon sensor histidine kinase	Phosphate regulation	P-starvation response regulation
K02039	*phoU*	Negative regulator of PhoR/PhoB two-component regulator	Phosphate regulation	P-starvation response regulation

**Table 2 microorganisms-12-02194-t002:** Basic physical and chemical properties of soil under different land use patterns (mean ± SE).

Soil Type	pH	EC (ms/cm)	TK (g/kg)	TN (g/kg)	AK (mg/kg)	TP (g/kg)	SOC (g/kg)	NH_4_^+^-N (mg/kg)
Tamarisk forest	7.6 ± 0.13 c	1.46 ± 0.35 b	8.64 ± 3.13 a	1.34 ± 0.31 a	353.56 ± 48.22 a	1.54 ± 0.23 a	2.6 ± 0.13 c	16.73 ± 0.29 a
White wax forest	8.3 ± 0.13 b	1.22 ± 0.85 b	6.89 ± 1.45 a	1.35 ± 0.18 a	311.30 ± 36.04 ab	1.25 ± 0.19 b	8.56 ± 0.31 b	12.06 ± 0.44 c
Farmland	7.59 ± 0.3 c	1.28 ± 0.53 b	3.83 ± 1.79 b	0.61 ± 0.14 b	265.72 ± 37.43 b	1.36 ± 0.05 ab	1.0 ± 0.18 d	11.90 ± 0.25 c
Wetlands	9.22 ± 0.4 a	3.35 ± 1.6 a	3.54 ± 2.32 b	0.87 ± 0.16 b	205.00 ± 51.33 c	1.58 ± 0.26 a	9.42 ± 0.24 a	15.09 ± 0.25 b
Grassland	9.13 ± 0.32 a	1.41 ± 0.22 b	3.41 ± 0.21 b	0.68 ± 0.29 b	276.34 ± 21.27 b	1.60 ± 0.15 a	2.86 ± 0.17 c	6.30 ± 0.23 d

Note: Different letters in the same column mean a significant difference at a 0.05 level. TK: total potassium; TN: total nitrogen; AK: available potassium; TP: total phosphorus; SOC: soil organic carbon.

## Data Availability

The data presented in this study are available on request from the corresponding author. Data available on request due to restrictions, e.g., privacy or ethics.

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
