# Peer review of "Metagenomic Analysis Reveals the Effects of Different Land Use Types on Functional Soil Phosphorus Cycling: A Case Study of the Yellow River Alluvial Plain"

_microorganisms, 2024, doi:10.3390/microorganisms12112194_

Round 1
Reviewer 1 Report
Comments and Suggestions for Authors
The article with the title “Metagenomic Analysis Reveals the Effects of Different Land Use Types on Functional Soil Phosphorus Cycling: A Case Study of the Yellow River alluvial plain” has not declared a clear aim „This study systematically examined the impact of different land use types”. How many types did you collect the samples? Insert it in the text „Soil Sample Collections. How much time did you monitored the different ecosystems? You collected once all samples, you did the analysis and that is it?
Please remove from the abstract the p value row 22, also you can remove rows 32-33, let this information in the conclusion section only. A general conclusion in the abstract is more than fine.
In the keywords you could add ecosystems. This study is comparing ecosystems like tamarisk forest, ash forest, wetland, farmland, and grassland.
The introduction section should highlight results for all type of ecosystems-land use type treated in this manuscript. Also could be improved.
At the end of the introduction section please state the aim of the study, the hypothesis and the information between rows 72-88 should be part of the material and methods section.
In the material and methods, please cite the methods used for each assessment that you did.
Please also be careful to formatting style, you have some missing spaces within the text please see row 126, row 79 and also check it in the entire text.
Rows 194-195 please rephrase the sentence and cite table 2.
Also add a note under table 2 to explain the statistical letters a significant threshold, make the table standing alone, note all abbreviation with entire name, tamarisk with uppercase
Rows 212-214 please rephrase the text highlight the main idea and cite the figure in brackets.
Rows 242-243 > The phosphate regulation gene cluster was most prevalent in grassland soils, significantly exceeding its abundance in other land-use types. What other? All others?
Make figure 3 larger.
Figure 3 caption missing statistical test name and abbreviation meaning the same for figure 4.
The second conclusion should be reformulated as a conclusion not as a result please see 451-455
Author Response
We are very grateful for the reviewers' and editor's comments. We will upload our responses to the revisions and comments in PDF format, with the revisions highlighted in yellow. Once again, thank you!

Reviewer 2 Report
Comments and Suggestions for Authors
Dear Editor and Authors,
The approach and topic of the study are highly relevant; however, there are some points that need improvement, as outlined below.
First, to establish a connection between P dynamics and its interaction with N and C, I recommend reviewing and discussing this topic more thoroughly.
Please include the hypothesis before the study's objectives.
The different management practices studied involve varying fertilizer inputs to meet the needs of each crop, which directly impacts the biogeochemical cycle of P. Therefore, the fertilization history of the areas should be provided, and this should become a central point in the discussion of the article. This is important because plants have different growth rates and export rates, affecting P dynamics.
Author Response
We are very grateful to the reviewers and editors for their review comments, and we will upload the responses on the revisions and the revised version of the manuscript in pdf form, where the revisions will be marked in yellow, thanks again!

Round 2
Reviewer 1 Report
Comments and Suggestions for Authors
Dear authors,
You made some changes, however not that satisfactory for the improvement of your manuscript so I will try to be clearer:
Please remove systematically from your entire manuscript
Add citation of all the methods used for the assessment of all parameters.
Author Response

(The authors gave the same response as above.)
